# The Effect of Foliar Application of Magnetic Water and Nano-Fertilizers on Phytochemical and Yield Characteristics of Fennel

Shahin Faridvand [1], Reza Amirnia [1,*], Mehdi Tajbakhsh [1], Hesham Ali El Enshasy [2,3,4] and R. Z. Sayyed [5]

[1] Department of Plant Production and Genetics, Faculty of Agriculture, Urmia University, Urmia 5756151818, Iran; qartal53@yahoo.com (S.F.); mtajbakhshurmia@gmail.com (M.T.)

[2] Institute of Bioproduct Development (IBD), Universiti Teknologi Malaysia (UTM), Skudai, Johor Bahru 81310, Malaysia; henshasy@ibd.utm.my

[3] School of Chemical and Energy Engineering, Universiti Teknologi Malaysia (UTM), Skudai, Johor Bahru 81310, Malaysia

[4] City of Scientific Research and Technology Applications (SRTA), New Burg Al Arab, Alexandria 21934, Egypt

[5] Department of Microbiology, PSGVP Mandal's Arts, Science and Commerce College, Shahada 425409, India; sayyedrz@gmail.com

\* Correspondence: r.amirnia@urmia.ac.ir or ramirnia@gmail.com

**Abstract:** Environmental factors, especially nutrients, can influence the production of medicinal plants. Thus, the present study assessed the response of some morphological and physiological characteristics of fennel ecotypes to the foliar application (magnetic water, organic and chemical fertilizers). The study was a factorial experiment based on the randomized complete block design, with three replications and 25 treatments at the research farm of the Agricultural and Natural Resources Research and Education Center of West Azerbaijan province, Iran, in the spring and summer of two consecutive years, 2014–2015 and 2015–2016. The first factor was assigned to fennel landraces (Gaziantep, Hamedan, Urmia, Yazd, and Shiraz) and the second factor to the foliar application (nitrogen nano-fertilizer, magnetic water, urea, chicken manure, and the control). Results showed that interaction of fertilizer treatment and landrace increased fresh and dry weight, biological yield, and seed yield significantly. In the first year, the highest fresh weight (166 g) and dry weight (35.5 g) were observed in the Gaziantep landrace fertilized with chicken manure. The highest anethole and fenchone contents (81.75% and 7.92%, respectively) were observed in the landraces treated with chicken manure. Based on the percentages, the Urmia landrace had the highest anethole percentage (83.2%), and the Shiraz landrace had the lowest one (77.5%). The highest fenchone contents (9.61%) and the lowest (2.18%) were observed in the Yazd and Urmia landraces. Due to the positive effect of application of chicken manure on improving the studied traits of fennel, it is recommended to include chicken manure inputs to enhance the efficiency of crops, reduce environmental pollution, and move toward sustainable agriculture.

**Keywords:** essential oil; chlorophyll; nutrient; anetol

## 1. Introduction

Medicinal plants are precious resources in Iran's broad landscape of natural resources, which can play an essential role in health, employment, and non-petroleum export if recognized, cultivated, developed, and exploited scientifically and correctly [1]. Fennel (*Foeniculum vulgare* Mill) is an ancient medicinal plant from the family Apiaceae, used as a raw material for different industrial medications [2]. This plant is cultivated and distributed in various Mediterranean regions and the Middle East, including Italy, Turkey, and Iran [3]. Fennel fruits are spindly shaped with narrow ends in green or light brown [4]. The whole body of this plant contains essential oil, but its fruits are the richest source of essential oil. The essential oils are accumulated through channel-shaped structures formed

by gland cells and distributed across the plant, and their active ingredients are used to heal cough, stomachache, flatulence, and dyspepsia in children and stimulate milk production in mothers [5]. The essential oil content of the fennel (3–5.5%) is greater than that of other species from the same family [6].

Given the limitation of water resources in Iran, the agricultural sector's use of unconventional water resources has been proposed as a solution. One of these methods is to pass water through a magnetic field before irrigation to improve water productivity [7]. Similar to other living organisms, plants are influenced by the Earth's magnetic field. The application of magnetic fields increases plant growth and development both in vitro and in field conditions. The practical application of magnetic fields has been reported to influence seed germination, seedling development positively, and yields of some plants, such as industrial crops [8,9], medicinal plants [10], fruits and vegetables [11], horticultural plants [12], and trees [13]. The effect of a magnetic field on plant growth and development depends on the magnetic field specifications, such as polarity, intensity, exposure duration, and magnetism type [14]. When water passes through a magnetic field, the hydrogen and Van der Waals bonds between its molecules break, and when the eponymous homonymous parts of water molecules are aligned, they will occupy less space, and consequently, their freedom and mobility will increase, and their surface tension will decrease [15]. In magnetic form, water fluidity increases, its wettability escalates, and it can be absorbed readily. Subsequently, more minerals are supplied to the plants, and their growth and yields are enhanced [16]. Mahmoud and Amira [17] reported that the grain weight, grain yield, and biological yield of bread wheat were higher when treated with 31%, 28%, or 25% magnetic water than when not. According to El Sayed [18], the plant height of fava beans was 27.39% greater in the non-magnetic water treatment.

Chemical fertilizers have played a significant role in meeting the nutrient requirements of plants and increasing crop yields in the last 30 years. However, the excessive application of them, e.g., urea, has brought about the contamination of groundwater resources, soils, and water resources and the over-accumulation of nitrogen in plant tissues in addition to increasing production costs [19,20]. Dahmardeh [21] found that although chemical fertilizers were valuable, production increased to a greater extent when chicken manure was applied. A primary goal of sustainable agriculture is to reduce chemical fertilizers and increase organic fertilizers, especially manures [22]. The application of manures, e.g., chicken manure, satisfy the nutrient requirements of plants and improves soil conditions, plant growth, and seed yield, and it can contribute to ecosystem sustainability in the long run. They can be an excellent supplement for chemical fertilizers if available in large quantities [23,24]. In this respect, chicken manure is rich in nutrients. It has other features, such as the gradual release of nitrogen (which can reduce nitrate leaching), gradual release of potassium and calcium compounds (which can be effective in adjusting soil pH), and high organic matter content (which can increase water and nutrient retention capacity) [24]. Azzaz et al. [25] revealed that the mixture of organic and biological fertilizers was most effective in increasing fennel plant growth, yield, and oil content. Ahmed and Bistgani et al. [26,27] reported that rose plants exhibited significant differences in their height, number of auxiliary branches, dry weight, and flower yield when treated with chicken manure versus the other fertilizers. Golestaneh et al. [28] found that cattle and chicken manure application to lavender plants changed some of their morphological traits, such as flower number, leaf, seed number, and 1000-seed weight significantly. A research study on safflower showed the significant effect of manure on the number of auxiliary branches, 1000-seed weight, and seed yield versus the control [29]. Al-Enzy et al. [30] found that manure increased most physiological traits of fennel. Moradi et al. [19] found that applying organic fertilizers changed anethole percentage significantly versus fenchone, estragole, and limonene. Similarly, Abbas and Abdul Muttalib [31] showed that anethole content was higher at higher N fertilizer levels, whereas fenchone content was higher at higher P fertilization levels.

A method to reduce the application of chemical fertilizers is to use nano-fertilizers to feed the plants [32]. Research on several plants has improved physiological activities, vegetative growth, protein level, and yield with nano-fertilization [33]. Abdelkader et al. [34] found that nano-fertilizers increased vegetative growth (height, the number of auxiliary branches, and dry weight), seed yield, essential oil yield, P content, and chlorophyll content of fennels. Nano-fertilizers improve nutrient use efficiency and alleviate the likely adverse effects of nutrient toxicity. Thus, nanotechnology has various aspects for accomplishing sustainable agriculture, especially in developed regions [35].

The present study aimed to cast light on the effect of different fertilization solutions and magnetic water on various indigenous and foreign landraces and their comparison with the local landrace (Urmia) of Fennel. An alternative can be found for nitrogen fertilizer to the environmental problems of its excessive application. Conversely, we aimed to compare the yield of different landraces to replace fennel in other cultures in the region to increase farm productivity by changing cropping patterns.

## 2. Materials and Methods

The research was conducted at the research farm (Long. $45°00'$ E, Lat. $37°40'$ N, elevation 1363 m from sea level) in the Agricultural and Natural Resources Research and Education Center of Western Azerbaijan province, Iran, to determine the effect of the foliar application of organic fertilizer (chicken manure), inorganic fertilizers (N nano-fertilizers and urea), and magnetic water on some morphological traits (fresh and dry weight), physiological traits (chlorophyll *a* and *b*, N, P, and K content, and essential oil yield), seed yield, and biological yield of fennel landraces.

Before the experiment initiation, the soil of the research farm was sampled from a depth of 0–30 cm to analyze its chemical and physical features; also, the physical and chemical characteristics of the chicken manure were measured. The results are presented in Tables 1 and 2. The latitude (degree, minute), longitude (degree, minute), and height (meter) of the cities whose landraces were used are presented in Table 3.

**Table 1.** The physical and chemical characteristics of the soil in the experimental farm.

| Year | Clay (%) | Silt (%) | Sand (%) | O.M (%) | pH | Available Nutrients (mg/kg$^{-1}$) | | | EC Ds/m | S.P (%) | TNV (%) |
|---|---|---|---|---|---|---|---|---|---|---|---|
| | | | | | | N | P | K | | | |
| First | 35.5 | 38 | 26.5 | 2 | 7.9 | 0.2 | 8.15 | 408 | 1.13 | 46 | 15.8 |
| Second | 34 | 38 | 28 | 1.47 | 7.88 | 0.23 | 7.25 | 416 | 1.82 | 47 | 15.1 |

**Table 2.** The physical and chemical characteristics of the chicken manure.

| Year | Cu (mg/kg$^{-1}$) | Zn (mg/kg$^{-1}$) | K (mg/kg$^{-1}$) | P (mg/kg$^{-1}$) | N (mg/kg$^{-1}$) | pH | C/N Ratio | O.M (%) | Moisture (%) |
|---|---|---|---|---|---|---|---|---|---|
| First | 106.8 | 400 | 20,000 | 17,500 | 50,000 | 7.6 | 12.8 | 34.1 | 28 |
| Second | 103 | 395 | 18,000 | 17,000 | 52,000 | 7.5 | 11.2 | 35 | 30 |

**Table 3.** The latitude, longitude, and height of the cities whose landraces were used.

| | Shiraz | Yazd | Urmia | Hamadan | Ghazian Tep |
|---|---|---|---|---|---|
| Longitude (Degrees, minutes) | $34°22'$ | $45°$ | $48°32'$ | $56°36'$ | $52°32'$ |
| Latitude (Degrees, minutes) | $29°36'$ | $33°22'$ | $34°5'$ | $34°4'$ | $46°4'$ |
| Height (meter) | 1519 | 1230 | 1363 | 1803 | 859 |

The soil of the study site was loam-sandy, and its climate was semi-arid. The first factor was assigned to fennel landraces at five levels (including Ghazian tep and the local landraces of Hamadan, Urmia, Yazd, and Shiraz), and the second factor was assigned to the foliar application at five levels (including N nano-fertilizers, magnetic water, urea, chicken manure, and control). Fertilizer treatments were applied by spraying on the plants. After

land preparation, the seeds were sown at a depth of 3 cm in plots with a length of 3 m on 8 rows with an inter-row spacing of 45 cm and an on-row spacing of 15 cm. The irrigation was performed by the sprinkler system every seven days, and the plots were regularly weeded. The fertilizers were applied at three stages—bolting initiation, early flowering stage or initiation bolting, and flowering flower.

The magnetic water solution was prepared at a magnetic strength of 200 mT with a Russian SIB8 water magnetization device (U050 mg, 0.5 inches, output 4–6 m$^3$/h, 30 mT L.C.C.) [36]. The nano-N fertilizer was prepared at a rate of 2:1000 as per the recipe on the label of 25% nano-N fertilizer procured from Pasargad Mehr Ragbarg Company, Tehran, Iran. An amount of 5 kg of urea fertilizer was dissolved in 100 L of water [37]. To produce the chicken manure solution, a mixture of water and chicken manure was prepared at a ratio of 1:10, kept at room temperature for 48 h, shaken, and filtered through two thin pieces of cloth [38]. In both cropping years, morphological traits (fresh and dry weight), physiological traits (chlorophyll *a* and *b*, N, P, and K contents, and essential oil yield), seed yield, biological yield, and harvest index of the fennel landraces were recorded. To calculate essential oil yield, essential oil percentage was multiplied by the yield of each treatment.

### 2.1. Measurement of Essential Oil Content

To determine essential oil content, 30 g of dried seeds from the plots were separately ground and extracted by the water-distillation method with a Clevenger for 3 h. Then, essential oil efficiency was calculated based on the sample's dry weight [39].

### 2.2. Measurement of Chlorophyll a and b Contents

To determine chlorophyll *a* and *b* contents, 0.2 g of fresh leaf tissue was gradually crashed with 5 mL of 80% acetone in a mortar until the chlorophyll was released into the acetone solution. Then, the solution volume was adjusted to 25 mL. It was then centrifuged at 4000 rpm for 10 min. The absorption of chlorophyll a and b were read with a spectrophotometer at 645 and 663 nm, respectively [40].

### 2.3. Measurement of Nitrogen Content

Nitrogen content was measured by the Kjeldahl method at two stages of digestion and distillation [41]. One gram of ground seeds was mixed with 5 g of catalyzer (copper sulfate, potassium sulfate, and copper oxide), and it was then added with 20 mL of 98% sulfuric acid. The samples were kept at 410 °C in a Kjeldahl (Gerhardt Company, Bonn, Germany) device for 1.5 h. After they were taken out of the device, 20 mL of distilled water was added, and then they were titrated with Titrasol sulfuric acid. The amount of acid applied in titration was placed in the following equation to yield nitrogen content.

### 2.4. Measurement of Potassium Content

Potassium content was determined by using a Sherwood 410 flame-photometer and the drawn standard curve [42]. To determine Fennel potassium content, 1 g of dry ground and meshed sample was placed in a furnace at 550 °C for 24 h. After digestion by dry burning method (with HCL), the samples were adjusted to 100 mL using distilled water. Using a Clinical pfp7 flame photometer (Jenway Gransmore Green Felsted, Dunmow Essex, CM6 3LB, ENGLAND, Model 8515), first potassium standards and then the main samples were read by the flame emission method. To prepare the standard, 9.53 g of potassium chloride was solved in water and its volume was adjusted in a 1 L volumetric flask (thick standard). Then, for a series of standard solutions, 10, 8, 6, 4, 2, and 0 mL of the thick standard was poured into 100 mL volumetric flasks containing 50 mL of water and 4.5 mL of thick sulfuric acid, and it was adjusted to the desired volume.

### 2.5. Measurement of Phosphorous Content

Phosphorous content was estimated based on the color intensity of the solutions using a Halo DB-20UV-VIS double-beam spectrophotometer at 470 mμ [43]. To measure Fennel

phosphorus content, 1 g of the sample was ground and meshed. After it was digested by dry burning (with HCL), it was adjusted to 100 mL by adding distilled water. Then, 5 mL of the sample was mixed with 5 mL of yellow solution (ammonium heptamolybdate + ammonium vanadate) and then its volume was increased to 25 mL by adding distilled water. After 0.5 h, the samples were filtered through a filter paper and the resulting extract was read at 470 nm with a spectrophotometer (Spectrophotometer APEL Model PD303UV, Kawaguchi City, Saitama, Japan). First, phosphorus standards and then the main samples were read. To prepare the standard, 2.19 g of KH2PO4 was first solved in a slight amount of water and was adjusted in volume in a 1-L volumetric flask (thick standard). For the series of standard solutions, 10, 8, 6, 4, 2, and 0 mL of the thick standard was taken, and 5 mL of zinc molybdate ammonium was added and adjusted to 25 mL.

### 2.6. Essential Oil Analysis

The compounds constituting seed essential oil were determined with a chromatograph and gas chromatography-mass spectrometry (GC/MS) in the laboratory of Urmia Academic Center for Education, Culture, and Research. The percentages of essential oil compounds were obtained based on the under-curve area of the chromatograph spectrum derived from GC [44]. Fennel EO was extracted by using the water distillation method. Briefly, 50 g of dry matter (leaf + flower) from each plot was weighed; briefly ground in 500 mL water, before boiling in a Clevenger for 3 h to extract the EO, which was then weighed. The EO content and EO yield were calculated as follows [45]: gas chromatography–mass spectrometry analysis was undertaken using an Agilent 7890/5975C (Santa Clara, CA, USA) GC/MSD. An HP-5MS capillary column (5% phenyl methyl polysiloxane, 30 m length, 0.25 mm id.0.25 μm film thicknesses) was used to separate the EO components. The following oven temperature was applied: 3 min at 80 °C, before increasing by 8 °C/min to 180 °C, and held for 10 min at 180 °C. Helium was used as the carrier gas at a flow rate of 1 mL/min. The sample was injected (1 μL) in split mode (1:50). The EI mode was 70 Ev. Mass range was set from 40 to 550 $m/z$. The components were recognized by comparing the calculated Kovats retention indices (RIs), from a mixture of n-alkaneseries (C8–C30, Supelco, Bellefonte, CA, USA) and mass spectra [44]. GC-FID analysis was undertaken with an Agilent 7890 an instrument. Separation was performed in an HP-5 capillary column. The analytical conditions were the same as above. Quantification methods were the same as those reported by Morshedloo et al. [46].

### 2.7. Statistical Analysis

The SAS (v9.4) software package analyzed data, and the means were compared by Duncan's multiple range test at the 1% and 5% levels.

## 3. Results and Discussion

Tables 4 and 5 show the analysis of variance related to the traits measured in this study.

### 3.1. Fresh and Dry Weight

Results in Table 6 show that the highest fresh weight (166 g) and dry weight (35 g) were obtained from the Ghaziantep plants treated with chicken manure in 2014, which increased 153% and 218%, respectively, compared to 2015, such that the lowest fresh weight (65.5 g) and dry weight (11 g) were related to the Ghaziantep plants in the control treatment in 2015. The significant increase in chicken manure's fresh and dry weight may be related to the extension of the growth period and the retardation of harvest time [47]. Organic manures are slow releaser and provide nutrients throughout growth period [48].

**Table 4.** Complete analysis of variance (mean squares) for the measured traits of sweet fennel.

| Sources of Variance | df | Oil Yield (kg/ha) | Biological Yield (kg/ha) | Seed Yield (kg/ha) | Dry Weight (g) | Fresh Weight (g) |
|---|---|---|---|---|---|---|
| Y | 1 | 224.4 ** | 26,779,317 ** | 3,734,179 ** | 124.2 ** | 2468 ** |
| R × Y | 2 | 29.07 ** | 1,612,841 ** | 4,9095 ** | 3.2 ns | 59.4 ns |
| L | 4 | 406.8 ** | 3,954,296 ** | 267,121 ** | 101.9 ** | 541.9 ** |
| T | 4 | 117.7 ** | 4,252,999 ** | 188,794 ** | 31.9 * | 407.8 ** |
| L × T | 16 | 44.52 ** | 2,127,966 ** | 9,6580 ** | 97.7 ** | 1439.5 ** |
| T × Y | 4 | 99.95 ** | 5,441,764 ** | 200,400 ** | 350.3 ** | 3234.1 ** |
| T × Y | 4 | 57.97 ** | 783,887 ns | 43,057 ** | 80.1 ** | 93.7 ** |
| L × T × Y | 16 | 45.88 ** | 2,123,625 ** | 113,983 ** | 75.2 ** | 979.9 ** |
| Error | 96 | 8.34 | 505,972 | 1,2469 | 3.9 | 61.3 |
| CV (%) | | 13.08 | 19.69 | 10.76 | 10.8 | 8.16 |

**, significant at α, 1%; *, significant at α, 5%; ns, non-significant; L, landrace; T, treatment; Y, year; R—replication; df—degree freedom; CV—coefficient of variation

**Table 5.** Complete analysis of variance (mean squares) for the measured traits of sweet fennel.

| Sources of Variance | Df | P (kg/ha) | K (kg/ha) | N (kg/ha) | Chlorophyll *a* (kg/ha) | Chlorophyll *b* (kg/ha) |
|---|---|---|---|---|---|---|
| Y | 1 | 188.1 ** | 1450.8 ** | 4996.3 ** | 666.1 ** | 13834 ** |
| R × Y | 2 | 1.2 ns | 15.1 ns | 72.7 ** | 48.1 * | 74.4 ns |
| L | 4 | 20.1 ** | 377.8 ** | 573.4 ** | 391.4 ** | 998.2 ** |
| T | 4 | 11.9 ** | 54.4 ** | 299.9 ** | 627.4 ** | 1706.6 ** |
| L × T | 16 | 6.8 ** | 112.3 ** | 183.5 ** | 462.6 ** | 778.5 ** |
| T × Y | 4 | 13.2 ** | 149.7 ** | 297.9 ** | 549.8 ** | 1531 ** |
| T × Y | 4 | 4.9 ** | 155.6 ** | 149 ** | 56.5 * | 222 ** |
| L × T × Y | 16 | 4.4 ** | 134.1 ** | 172.9 ** | 218.7 ** | 675.9 ** |
| Error | 96 | 0.71 | 5.03 | 15.5 | 19.3 | 36.4 |
| CV (%) | | 13.8 | 11.5 | 11.4 | 13.1 | 10.9 |

**, significant at α, 1%; *, significant at α, 5%; ns, non-significant; L, landrace; T, treatment; Y, year.

**Table 6.** The effect of different sources of fertilizer and magnetic water on plant dry weight and fresh weight of sweet fennel in 2014–2015 and 2015–2016 years.

| Treatments | Dry Weight Ecotype | | | | | Fresh Weight Ecotype | | | | |
|---|---|---|---|---|---|---|---|---|---|---|
| | Qaziantep | Hamadan | Urmia | Yazd | Shiraz | Qaziantep | Hamadan | Urmia | Yazd | Shiraz |
| **2014** | | | | | | | | | | |
| Nano Nitrogen | 100 e–m | 121.6 cd | 104 d–k | 99 h–p | 91 m–p | 26.7 de | 23.9 d–h | 18.5 j–n | 16 n–p | 22.9 e–j |
| Magnetic Water | 126 bc | 71.4 p–s | 143 b | 62 j–q | 87 j–q | 31 bc | 12.6 o–t | 17 i–n | 11 m–o | 21.4 f–l |
| Urea | 100 e–m | 95.4 g–n | 117 c–e | 85 e–m | 100 e–m | 23.8 d–h | 14.7 n–r | 21.5 f–l | 19 r–t | 21.8 f–k |
| Chicken Manure | 166 a | 116.3 c–f | 70 q–s | 91 s | 89 i–q | 35.5 a | 20.7 g–m | 18.4 j–n | 20 i–n | 21 f–l |
| Control | 91 h–p | 76.3 n–s | 110 c–h | 87 k–r | 95 g–n | 24 d–h | 15.3 n–q | 23.1 d–i | 23 h–m | 16 m–o |
| Main B | 116.6 | 96.2 | 108.8 | 84.8 | 92.4 | 28.4 | 17.5 | 19.7 | 17.8 | 20.6 |
| **2015** | | | | | | | | | | |
| Nano Nitrogen | 80 m–s | 77 n–s | 82.5 l–r | 101 e–l | 109 c–i | 18.3 k–n | 14.3 n–s | 18.6 i–n | 25 d–f | 28 b–d |
| Magnetic Water | 83 l–r | 91 h–p | 92.5 e–m | 122 c–d | 81 l–s | 15.2 n–q | 20.8 g–m | 22 f–k | 27 c–e | 9.4 t |
| Urea | 77 n–s | 92 h–o | 96.7 f–n | 126 c | 81 l–s | 15.5 n–p | 20.7 g–m | 24 d–h | 24 d–h | 16 m–o |
| Chicken Manure | 83.5 k–r | 113 c–g | 89.4 h–q | 87 j–r | 74 o–s | 14.3 n–s | 31.3 b | 11.4 p–t | 14 n–s | 12 o–t |
| Control | 65.6 r–s | 106 d–g | 101 e–l | 77 n–s | 70 q–s | 11 q–t | 25.2 d–g | 22 f–k | 21 f–l | 10.2 st |
| Main B | 77.8 | 95.8 | 72.5 | 103 | 83 | 14.9 | 22.53 | 19.6 | 21.9 | 15.2 |

Similar letters above the values show non-significant difference at *p* < 0.05 by Duncan test.

Furthermore, chicken manure is released gradually and satisfies the nutrient requirements throughout the growth period. The interactions of the factors (Table 6) reveal that the dry and fresh weights of the fennels were influenced by time.

Zhou et al. [49] reported that the vegetation growth of radish and pakchoi luffa plants increased when treated with organic fertilizers versus chemical fertilizers. These fertilizers are rich in nitrogen, increasing photosynthesis rate, increasing fresh and dry weight [50,51]. Organic manures would have provided micronutrients such as Fe, Zn, Cu, Mn and Mg at an optimum level. Copper and manganese are important coenzymes for certain respiratory reactions. Iron is involved in the chlorophyll synthesis pathway. Zinc is involved in the biochemical synthesis of most important phytohormones. Magnesium is involved in chlorophyll synthesis, which in turn increases the rate of photosynthesis [52]. The promoting effect of organic manure on growth of fennel plants may be attributed to the role of organic fertilizer in physiological and biological process as a source of N, P, S and contains high content of B and Mo [53].

Shabrangi and Majd [54] reported that the increases in biomass and fresh weight required metabolic changes, especially in protein biosynthesis. Abdul Qados and Hozayn [16] ascribed the effect of magnetic water on growth to the stimulation of cell metabolism and mitosis. According to Olowoake and Adeoye [55], the effect of cattle manure on corn was similar to the effect of chemical fertilizer. The fresh weight of plants may be related to the high nutrient content of organic fertilizers [56]. These findings are consistent with the literature reports of [57–59].

### 3.2. Fruit Yield and Biological Yield

The results in Table 7 indicate that the different sources of foliar application had significant effects on seed and biological yields in both years. The highest seed yield (2039 kg/ha) and biological yield (7604 kg/ha) were obtained from the Ghaziantep landrace treated with magnetic water in the first year, which increases 235% and 272% respectively compared to 2015. Conversely, the lowest seed yield (608 kg/ha) was observed in Ghaziantep treated with urea, and the lowest biological yield (2040 kg/ha) was observed in Yazd treated with the control, both in the second year. These results show improvements in photosynthesis, hormone balance, growth parameters, and Transmission efficiency translocation efficiency of the plants treated with magnetic water [17,60]. These findings are consistent with the literature reports [61–67] respective to green cumin, fennel, celery, and black cumin.

**Table 7.** The effect of different sources of fertilizer and magnetic water on fruit yield and biological yield of sweet fennel in 2014–2015 and 2015–2016 cropping years.

| Treatments | Fruit Yield (kg/ha) Ecotype | | | | | Biological Yield (kg/ha) Ecotype | | | | |
|---|---|---|---|---|---|---|---|---|---|---|
| | Qaziantep | Hamadan | Urmia | Yazd | Shiraz | Qaziantep | Hamadan | Urmia | Yazd | Shiraz |
| **2014** | | | | | | | | | | |
| Nano Nitrogen | 1728 [b] | 1369 [cd] | 1067 [e–n] | 1317 [c–f] | 1227 [c–g] | 5003 [b–c] | 4848 [b–i] | 3234 [c–j] | 4454 [b–f] | 4364 [b–j] |
| Magnetic Water | 2039 [a] | 1118 [dk] | 952 [j–o] | 1051 [f–o] | 1303 [c–e] | 7604 [a] | 4068 [b–i] | 2785 [h–j] | 3870 [c–j] | 4765 [b–e] |
| Urea | 892 [k–r] | 1426 [c] | 1082 [e–n] | 914 [k–r] | 1340 [c–i] | 2438 [h–j] | 5719 [b] | 3492 [c–j] | 2111 [g–j] | 3326 [b–e] |
| Chicken Manure | 1294 [c–h] | 1050 [e–o] | 1091 [d–m] | 830 [k–r] | 1283 [c–f] | 4845 [b–d] | 3586 [c–j] | 2412 [h–j] | 2745 [f–g] | 4454 [b–f] |
| Control | 1048 [e–n] | 1231 [c–j] | 1013 [h–q] | 876 [k–r] | 1318 [g–p] | 3477 [c–j] | 3942 [b–i] | 3543 [c–j] | 2888 [e–j] | 5001 [b–c] |
| Main B | 1400 | 1239 | 1041 | 998 | 1294 | 3072 | 4433 | 3093 | 3214 | 4382 |
| **2015** | | | | | | | | | | |
| Nano Nitrogen | 1029 [g–p] | 819 [i–r] | 860 [k–r] | 886 [k–r] | 968 [j–q] | 2727 [f–j] | 3345 [c–j] | 3399 [c–j] | 4145 [b–h] | 3157 [c–j] |
| Magnetic Water | 874 [k–r] | 1104 [d–f] | 857 [k–r] | 899 [k–r] | 975 [j–q] | 3468 [c–j] | 3088 [d–j] | 3632 [c–j] | 3598 [c–j] | 3630 [c–j] |
| Urea | 608 [k–r] | 914 [k–r] | 822 [i–r] | 998 [i–q] | 765 [q–r] | 4092 [b–r] | 3642 [c–j] | 2437 [h–j] | 3930 [b–j] | 3222 [c–j] |
| Chicken Manure | 1016 [h–q] | 875 [k–r] | 1028 [g–p] | 737 [p–r] | 634 [r] | 3241 [c–j] | 2275 [h–j] | 3579 [c–j] | 3083 [d–j] | 2235 [i–j] |
| Control | 788 [n–r] | 796 [m–r] | 907 [k–r] | 725 [q–r] | 813 [l–r] | 2889 [e–j] | 2763 [f–j] | 2917 [c–j] | 2040 [j] | 2861 [f–j] |
| Main B | 863 | 902 | 895 | 849 | 831 | 3283 | 3023 | 3193 | 3359 | 3021 |

Similar letters above the values show non-significant difference at $p < 0.05$ by Duncan test.

Hozayn and Qados [68] found that magnetic water improved stomata conductance and nutrient assimilation. Magnetic water reportedly has several physical and chemical

characteristics, such as polarity, surface tension, pH, electrical conductivity, and solubility [69], improving plant growth.

The higher seed yield in the first year may be associated with the optimal temperature. Since the Ghaziantep landrace outperformed the other studied landraces in growth parameters, it can be inferred that it had a higher seed yield, with the literature reports [70]. El-Abd et al. [71] emphasized that Spraying at the time of inflorescence foliar application during inflorescence had a significant effect on increasing fennel yield. The foliar application of mineral nutrients offers a method of supplying nutrients to higher plants that are more efficiently than methods involving root application when soil conditions are not suitable for nutrients availability [72]. Flower yield in the first year was greater than that in the second year [73].

### 3.3. Chlorophyll a and b

According to Table 8, the chlorophyll *a* and *b* contents of the fresh leaves of fennel were significantly influenced by foliar application. The Ghaziantep landrace treated with magnetic water in the first year exhibited the highest chlorophyll-*a* (116.2 kg/ha) and chlorophyll *b* (65.5 kg/ha) contents while the Yazd landrace treated with control in the second year exhibited the lowest chlorophyll *a* content (23.8 kg/ha) and the Shiraz landrace treated with chicken manure in the second year exhibited the lowest chlorophyll *b* content (18 kg/ha). Chloroplast has paramagnetic properties [74]. The magnetic movement of ions in a magnetic field is toward the downside of the field. The magnetic field has a remarkable impact on plants such that it increases energy. Chlorophyll enhancement can be attributed to increased ion fluidity and uptake in the magnetic field, which increases chlorophyll pigments, chlorophyll activity, translocation efficiency, and photo assimilation in the plant [75,76]. Sadeghipour and Aghaei [77] found that irrigation with magnetic water increased leaf area and photosynthesis rate, resulting in more assimilates for vegetative growth, which the increase in chlorophyll content can explain. Chloroplast contains pigments whose structure and position are affected by external factors [78]. The magnetic field makes oriented chloroplast and its elements directional. The same phenomenon is observed in solid objects (ions) and changes the absorption intensity. When a plant is exposed to an external magnetic field, the energy of its paramagnetic materials is increased, and this can activate plant hormones [79]. Research shows that a magnetic field influences photosynthesis mainly through increasing chlorophyll, not other plant carotenoids [80]. After the treatment with the magnetic field, the difference in the interception of short-wavelength waves was clearly shown with chlorophyll *b*. The treatment with the magnetic field influenced chlorophyll, thereby increasing the quality and germination of non-standard seeds [81]. This response can be explained by the presence of some paramagnetic elements in the chloroplast. The magnetic field decreases the effect of carotenoids and increases the share of chlorophyll *b* in absorption. To study the biochemical changes in chlorophyll under the effect of a magnetic field, Racuciu et al. [80] measured all carotenoids and nucleic acids involved in chlorophyll formation and found that the amount of nucleic acids was increased in the treatments with a low-power magnetic field. However, it was not observed in the treatments with a high-power magnetic field. In another study, Aladjadjiyan [79] explored the effect of magnetic field on the absorption of light spectrum by the photosynthesis system in several perennial ornamental plants and showed that the magnetic field changed the spectral absorption in all samples in both domains of absorption by pigments. Hence, it directly affected the photosynthesis system.

**Table 8.** The effect of different sources of fertilizer and magnetic water on chlorophyll *a* and *b* of sweet fennel plants in 2014–2015 and 2015–2016 cropping years.

| Treatments | Chlorophyll *a* Ecotype | | | | | Chlorophyll *b* Ecotype | | | | |
|---|---|---|---|---|---|---|---|---|---|---|
| | Qaziantep | Hamadan | Urmia | Yazd | Shiraz | Qaziantep | Hamadan | Urmia | Yazd | Shiraz |
| **2014** | | | | | | | | | | |
| Nano Nitrogen | 67 [e–i] | 79.6 [b–d] | 52.1 [j–o] | 78 [b–e] | 72 [c–g] | 40.2 [e–i] | 47.6 [de] | 27.8 [j–o] | 54 [cd] | 29 [m–r] |
| Magnetic Water | 116 [a] | 71.8 [c–g] | 44.7 [k–p] | 70 [d–h] | 76.4 [c–e] | 65.6 [a] | 40.4 [e–t] | 24.3 [i–n] | 38 [f–l] | 31 [i–r] |
| Urea | 33 [p–u] | 109 [a] | 58.6 [h–j] | 40 [o–s] | 81.6 [bc] | 20 [s–u] | 57.6 [bc] | 38.2 [f–j] | 18 [u] | 37 [f–m] |
| Chicken Manure | 73 [c–f] | 56.3 [ik] | 32.3 [q–u] | 36 [p–t] | 68.6 [d–h] | 42.3 [e–g] | 41.1 [e–i] | 19.5 [tu] | 33 [iq] | 21 [f–l] |
| Control | 55 [il] | 62 [f–j] | 58.5 [h–j] | 43 [m–r] | 87.5 [b] | 26.9 [o–t] | 23.8 [r–u] | 27.5 [n–t] | 30 [k–r] | 33 [n–p] |
| Main B | 68.8 | 62.5 | 49.3 | 53.4 | 77.4 | 42.1 | 42.5 | 27.5 | 34.6 | 30.2 |
| **2015** | | | | | | | | | | |
| Nano Nitrogen | 31 [r–u] | 51.7 [j–o] | 53 [jn] | 50.6 [i–o] | 43.5 [l–q] | 23.1 [r–u] | 39.2 [c–j] | 33.9 [g–o] | 63 [ab] | 22 [r–u] |
| Magnetic Water | 53.7 [j–m] | 44.7 [k–p] | 58.3 [h–j] | 59 [h–j] | 51.5 [j–o] | 39.3 [e–j] | 29.1 [m–r] | 36.7 [f–m] | 40 [e–i] | 26 [o–u] |
| Urea | 58.5 [h–j] | 60.8 [g–j] | 33.2 [p–u] | 58.7 [h–j] | 52 [j–o] | 37.9 [f–k] | 38.2 [f–j] | 27.5 [n–t] | 36 [g–n] | 29 [l–r] |
| Chicken Manure | 41.3 [n–s] | 27.3 [tu] | 54.5 [i–l] | 38.9 [p–s] | 30.5 [s–u] | 28.6 [m–s] | 23.9 [r–u] | 38.2 [f–j] | 42 [e–h] | 18 [u] |
| Control | 41.3 [n–s] | 40.3 [o–s] | 40.4 [o–s] | 23.8 [u] | 42.6 [m–r] | 24.9 [p–u] | 19.9 [tu] | 23 [r–u] | 20 [tu] | 25 [p–u] |
| Main B | 45.2 | 45 | 47.9 | 46.2 | 44 | 30.8 | 30 | 31.9 | 40 | 24 |

Similar letters above the values show non-significant difference at $p < 0.05$ by Duncan test.

### 3.4. Essential Oil Yield

The highest essential oil yield (38.8 kg/ha) was obtained from Ghazian tep treated with magnetic water in the first year, and the lowest (12.3 kg/ha) from Hamadan treated with control in the second year (Table 9). It is observed that the maximum essential oil yield was three times as great as the minimum one. The foliar application of manure and chemical fertilizer increased this trait versus the control significantly (Table 9). The local landraces of fennel differed significantly in essential oil yield such that the mean essential oil yield of Urmia in two years (21.87 kg/ha) was 38% higher than that of Hamadan (15.95 kg/ha). Although Hamadan had a higher seed yield (Table 9), its essential oil yield was lower than the other landraces due to its lower essential oil percentage. Other landraces had significant differences from Hamadan (Table 6). As is evident in Table 9, the essential oil yield in the first year was 15% higher than that in the second year (23.9 kg/ha versus 20.8 kg/ha). Since nutrients are the raw material for essential oil production, as they increase, leaf photosynthesis increases, resulting in the production of essential oil per unit area. Mohsen and Kassem [82] reported that magnetic water treatment increased essential oil significantly compared to the control due to magnetic water's higher water and nutrient uptake.

**Table 9.** The effect of some different sources of fertilizer and magnetic water on nitrogen content and essential oil of sweet fennel plants in 2014–2015 and 2015–2016 cropping years.

| Treatments | Nitrogen Content Ecotype (mg/g) | | | | | Essential Oil Ecotype | | | | |
|---|---|---|---|---|---|---|---|---|---|---|
| | Qaziantep | Hamadan | Urmia | Yazd | Shiraz | Qaziantep | Hamadan | Urmia | Yazd | Shiraz |
| **2014** | | | | | | | | | | |
| Nanonitrogen | 39 [a] | 34.4 [b–g] | 34.4 [b–g] | 33.3 [c–j] | 32.9 [d–j] | 34 [h] | 19.8 [g–k] | 24.7 [c–f] | 27 [cd] | 23 [d–g] |
| Magnetic Water | 38.3 [a] | 32.8 [e–j] | 32.4 [f–k] | 32.5 | 31.4 [i–w] | 38.8 [a] | 14.7 [l–o] | 23.3 [d–g] | 21 [f–j] | 23 [d–g] |
| Urea | 32.4 [f–k] | 34 [b–h] | 33.2 [c–j] | 31.9 | 33.1 [d–j] | 17.3 [f–n] | 19.8 [g–k] | 25.7 [c–e] | 18 [g–m] | 26 [c–e] |
| Chicken Manure | 34.2 [b–h] | 33 [d–j] | 34.2 [b–h] | 33.1 | 31 [j–n] | 26 [g–k] | 14.8 [m–o] | 25.7 [c–e] | 17 [j–o] | 29 [c] |
| Control | 33.5 [c–i] | 33 [c–j] | 28.7 [o] | 31.4 | 33 [e–j] | 22 [e–i] | 17.6 [i–n] | 25 [c–f] | 17 [j–o] | 19 [g–h] |
| Main B | 35.5 | 33.2 | 32.6 | 32.4 | 32.3 | 27.6 | 17.3 | 24.9 | 20 | 24 |

**Table 9.** *Cont.*

| Treatments | Nitrogen Content Ecotype (mg/g) | | | | | Essential Oil Ecotype | | | | |
|---|---|---|---|---|---|---|---|---|---|---|
| | Qaziantep | Hamadan | Urmia | Yazd | Shiraz | Qaziantep | Hamadan | Urmia | Yazd | Shiraz |
| | | | | | **2015** | | | | | |
| Nano Nitrogen | 33.6 [h–p] | 26.6 [o–s] | 25.1 [q–s] | 28.6 [m–s] | 30 [k–r] | 16 [h–o] | 12.33 [n–o] | 18 [h–n] | 20 [g–k] | 20 [g–k] |
| Magnetic Water | 30.7 [j–r] | 38.5 [e–j] | 26.9 [n–s] | 27.2 [n–s] | 29 [l–s] | 12.3 [a] | 17.3 [i–n] | 19.3 [g–l] | 19 [g–l] | 17 [i–n] |
| Urea | 29.7 [k–r] | 32.8 [i–q] | 26.7 [n–s] | 35.4 [hm] | 24.8 [rs] | 17.8 [h–n] | 16.7 [j–p] | 18 [h–n] | 23 [d–g] | 17 [i–n] |
| Chicken Manure | 34.1 [h–o] | 30.3 [k–r] | 34.1 [h–o] | 25.1 [q–s] | 21.3 [s] | 16.3 [j–o] | 13.3 [n–o] | 16 [k–o] | 16 [k–o] | 19 [g–h] |
| Control | 26.9 [n–s] | 25.9 [p–s] | 26.9 [n–s] | 21.8 [s] | 25.2 [q–s] | 17.3 [i–n] | 12.3 [p] | 22.3 [e–h] | 16 [k–o] | 19 [g–h] |
| Main B | 31 | 30.8 | 27.9 | 27.6 | 26 | 15.9 | 14.4 | 18.7 | 18.8 | 18.4 |

Similar letters above the values show non-significant difference at $p < 0.05$ by Duncan test.

### 3.5. The N Content of Fennel Fruits

The foliar application of the organic fertilizer, inorganic fertilizer, and magnetic water brought about significant differences in the N content of fennels (Table 9). The magnetic water increased the seed N content significantly versus the other solutions in both years. The highest N content of 39 mg/g was related to Ghazian tep treated with magnetic water in the first year, and the lowest of 21.3 kg/ha was related to the Shiraz landrace treated with chicken manure in the second year (Table 9). The increase in N under the magnetic water treatment can be related to the hydrogen bonds of water molecules, which are severely influenced by the magnetic field such that solubility increases. In addition, the magnetic field affected the cell membrane permeability of fava beans and changed the ion movement of the membrane [83]. Since magnetic water has different physical characteristics from conventional water, it has a higher capacity for the dissolution of salt and minerals [84]. Esitken and Turan [85] observed that the magnetic field increased the N content of strawberry leaves. The effectiveness of magnetic water varies with plant organs [86] and species [87]. The accumulation of ions positively affects plant growth and production, and magnetic fields play a key role in increasing the uptake capacity of ions and non-fluid ions.

Ions and nutrients are made directional by the magnetic field treatment. This phenomenon is similar to the effect of magnetic fields on solid objects. This changes the uptake rate, and obviously, plants that absorb more nutrients exhibit a higher rate of growth and higher yields [88]. Influenced by an external magnetic field, the energy of paramagnetic materials in plants, which is transported by free electrons, increases, resulting in the activation of plant hormones [80]. This finding is supported by the reports of Nasreen et al. [89] and El Sagan and Baset [90] about onion and Darzi [91] about parsley. The N content of pear seedlings and tomato seedlings irrigated with magnetic water was higher than conventional water [58,91]. Mohsen and Kassem [82] reported that magnetic water increased the N content of fennel fruits.

### 3.6. The P Content of Fennel Fruits

Based on the results of two-year combined variance analysis of the data, the interactive effect of the three factors was significant such that the maximum *P* content of 8.7mg/g was obtained from Ghaziantep treated with the magnetic water in the first year and the minimum one of 4.1 mg/g from the Shiraz landrace treated with the magnetic water in the second year (Table 10). Magnetic water changes soil's physical and chemical characteristics, increasing nutrient availability to plants [92]. Furthermore, it affects plant hormones, thereby increasing cell activity and improving plant growth [64]. The irrigation of pear and tomato seedlings with magnetic water increased their N content versus the application of conventional water [50,83]. In a study on chickpea and cowpea, Maheshwari and Grewal [64] reported the favorable effect of magnetic water on plant minerals. Mohsen and Kassem [82] reported similar results about the increased content of P in the seeds of fennel plants that were treated with magnetic water.

**Table 10.** The effect of different sources of fertilizer and magnetic water on potassium and phosphor contents of sweet fennel plants in the 2014–2015 and 2015–2016 cropping years.

| Treatments | Potassium Ecotype (mg/g) | | | | | Phosphor Ecotype (mg/g) | | | | |
|---|---|---|---|---|---|---|---|---|---|---|
| | Qaziantep | Hamadan | Urmia | Yazd | Shiraz | Qaziantep | Hamadan | Urmia | Yazd | Shiraz |
| **2014** | | | | | | | | | | |
| Nano Nitrogen | 27.8 a | 18.8 h–l | 17.4 l–r | 16.8 n–t | 17 o–t | 7.7 ab | 7.8 a | 5.8 h–o | 5.1 n–t | 5.7 g–q |
| Magnetic Water | 28.5 a | 19.2 g–j | 12 ef | 19.2 gk | 19.4 g–j | 8.7 j–q | 6.6 c–g | 7 bcd | 5.6 d–h | 5.4 m–t |
| Urea | 15.2 t–w | 19.4 g–j | 20 fh | 16.2 q–u | 18.5 h–l | 6.5 c–i | 5.4 m–t | 6 g–m | 5.3 n–t | 5.4 m–t |
| Chicken Manure | 22.4 de | 17.4 l–k | 14.5 vw | 17.2 l–s | 18.4 i–n | 4.3 abc | 5.2 o–t | 6.2 f–k | 6.5 d–h | 5.5 l–s |
| Control | 17.8 j–o | 17.2 l–s | 15.8 p–u | 18.3 j–n | 18 j–o | 4.7 t–v | 5.4 m–t | 4.9 r–h | 6.7 l–s | 6.6 c–g |
| Main B | 22.34 | 18.4 | 15.9 | 17.54 | 18.24 | 6.38 | 6.1 | 5.98 | 5.84 | 5.72 |
| **2015** | | | | | | | | | | |
| Nano Nitrogen | 18 h–o | 16.8 m–t | 18.1 h–m | 16.3 p–u | 19.4 g–j | 5.9 h–n | 6.9 b–e | 5.9 h–n | 5.7 k–r | 4.9 s–u |
| Magnetic Water | 15 q–u | 15 q–u | 17.9 j–p | 17.4 l–r | 19.5 f–g | 7.4 ab | 5.1 g–u | 6.5 d–i | 5.1 p–u | 4.1 u–v |
| Urea | 17.8 j–o | 17.8 j–o | 14.2 s–u | 18 j–o | 19.9 f–i | 6.6 c–g | 4 v | 6.3 e–j | 4.5 u–v | 5.7 j–q |
| Chicken Manure | 14.6 vw | 14.6 vw | 20.5 f–g | 14.1 w | 20 fh | 5.8 i–p | 4.7 t–v | 6.9 b–f | 5.7 j–g | 7.3 abc |
| Control | 15.8 o–r | 15.8 p–u | 14.4 vw | 15.7 o–r | 15.5 p–u | 5.4 m–s | 6.1 g–l | 7 b–d | 5.6 k–r | 4.5 w |
| Main B | 16.22 | 16 | 17.02 | 16.3 | 18.86 | 6.22 | 5.36 | 6.52 | 5.32 | 5.3 |

Same letters above numbers show non-significant difference at $p < 0.05$ by Duncan test.

### 3.7. The K Content of Fennel Fruits

The comparison of the two-year means revealed that the interaction of the three factors was significant on this trait. The Ghaziantep plants treated with magnetic water in the first year exhibited the highest K content of 28.5 mg/g and the Yazd plants treated with chicken manure in the second year exhibited the lowest one of 14.1 mg/g (Table 10). Magnetic fields make cell membranes permeable and change ion movements across the membrane [93]. This, eventually, increases osmotic pressure. The osmotic pressure of the cell membrane reaches a balance with an increase in K content and an increase in the K/Na ratio [94]. Researchers suggest that the accumulation of some minerals, e.g., K, in plants contributes to preserving cell turgor and improving the morphological and physiological traits of plants by preventing the degradation of cell walls against reactive oxygen species, increasing the activity of antioxidant enzymes, and increasing water use efficiency [95]. Esitken and Turan [85] showed that the magnetic field increased the K ion content of strawberries. The positive effect of magnetic water was reported on the K content of plants [57,66,82,96–98] for basil, faba, fennels, chickpea, and periwinkle. Hasan et al. [99] reported that applying magnetic water solution increased K uptake by increasing the K/Na ratio.

### 3.8. Essential Oil Composition

The comparison of the means indicated that the foliar application influenced essential oil composition significantly. The highest p-anisaldehyde, limonene, and alpha-pinene contents (3.28%, 4.41%, and 0.58%, respectively) were observed in the landraces treated with the magnetic water, and the highest anethole and fenchone contents (80.16% and 7.92%, respectively) were observed in the landraces treated with the chicken manure (Table 11). The highest alpha-pinene, beta-mercine, p-simene and beta-osimene contents (0.58%, 0.13%, 0.35% and 0.27%, respectively) were observed in the landraces treated with the magnetic water.

Abdul-Jaleel et al. [100] state that the concentration of secondary compounds in plants depends on environmental factors, such as nutrients and light, which influence plant yields and the quantity and quality of compounds. The main constituents of fennel include anethole, fenchone, limonene, estragole, and anise aldehyde. Darzi et al. [101] state that the high anethole content of fennel's essential oil improves its quality. Since the essential oil is a terpenoid compound whose constituents urgently need ATP and the presence of such elements as nitrogen is necessary [89], N availability increases anethole content and decreases estragole content. Jamshidi et al. [20] observed that applying chemical and

organic fertilizers increased anethole, fenchone, and limonene contents in fennels. The fenchone content was higher in most foliar application treatments than in the control treatment. Younesian et al. [102] revealed that the anethole percentage of fennel was higher than the other compounds and was affected by the fertilization treatments such that manure was most influential on anethole content. Based on the percentages, the Urmia landrace had the highest anethole percentage (83.18%), and the Shiraz landrace had the lowest one (77.52%) (Table 12). Baydar et al. [103] and Bettaieb et al. [104] argue that medicinal herbs' essential oil yield and quality depend on genotype, but they are influenced by climatic factors, soil physical and chemical conditions, and available nutrients. Some researchers also suggest that the quality of essential oil constituents is mainly affected by genotype, but environmental factors are less influential [105].

**Table 11.** The effect of different sources of fertilizer and magnetic water on the essential oil of fennel in the 2014–2015 and 2015–2016 cropping years.

| | Foliar Application | | | | |
|---|---|---|---|---|---|
| | Control | Chicken Manure | Urea | Magnetic Water | Nano-N |
| Trans-anetol | 79.2 | 80.16 | 78.9 | 78.98 | 81.21 |
| Fenchone | 7.32 | 7.92 | 6.89 | 7.37 | 7.56 |
| Limonene | 3.98 | 3.94 | 4.11 | 4.41 | 3.45 |
| Estragol | 4.21 | 4.67 | 4.52 | 4.96 | 4.68 |
| P-anisaldehyde | 2.35 | 2.77 | 2.76 | 3.28 | 2.96 |
| Alpha-pinene | 0.48 | 0.48 | 0.48 | 0.58 | 0.44 |
| Sabinene | 0.08 | 0.08 | 0.14 | 0.11 | 0.11 |
| Beta-mercine | 0.1 | 0.09 | 0.12 | 0.13 | 0.08 |
| p-simene | 0.09 | 0.11 | 0.1 | 0.35 | 0.1 |
| 1.8 Cineol | 0.15 | 0.22 | 0.16 | 0.2 | 0.18 |
| Camphor | 0.07 | 0.1 | 0.07 | 0.1 | 0.12 |
| Beta-osimene | 0.24 | 0.25 | 0.22 | 0.27 | 0.22 |
| Camphen | 0.01 | 0.01 | - | - | - |
| L-phelendrel | 0.01 | - | - | - | - |

**Table 12.** The essential oil of some fennel landraces in the 2014–2015 and 2015–2016 cropping years.

| | Landrace | | | | |
|---|---|---|---|---|---|
| | Shiraz | Yazd | Urmia | Hamadan | Qazian Tep |
| Trans-anetol | 77.52 | 80.82 | 83.18 | 78.68 | 78.04 |
| Fenchone | 8.75 | 9.61 | 2.18 | 8.58 | 8.53 |
| Limonene | 3.93 | 3.85 | 4.2 | 3.96 | 4.18 |
| Estragol | 5.27 | 4.31 | 5.27 | 3.87 | 4.18 |
| P-anisaldehyde | 3.04 | 3.25 | 2.91 | 2.12 | 2.61 |
| Alpha-pinene | 0.54 | 0.46 | 0.33 | 0.53 | 0.6 |
| Sabinene | 0.12 | 0.1 | 0.11 | 0.06 | 0.11 |
| Beta-mercine | 0.14 | 0.09 | 0.01 | 0.14 | 0.12 |
| P-simene | 0.09 | 0.09 | 0.15 | 0.05 | 0.08 |
| 1.8 Cineol | 0.21 | 0.17 | 0.16 | 0.15 | 0.12 |
| Camphor | 0.12 | 0.09 | 0.01 | 0.15 | 0.12 |
| Beta-osimene | 0.26 | 0.27 | 0.27 | 0.27 | 0.14 |
| Camphen | 0.01 | 0.01 | - | - | - |
| L-phelendrel | - | - | - | - | 0.01 |

Limonene percentage was lower in the treatments that had a higher anethole percentage. Most research studies have reported that applying organic and chemical fertilizers increases some compounds and decreases others [20,44,106–108].

## 4. Conclusions

The results showed that optimal yields could be obtained by identifying plant cultivars and through proper nutritional regimes. One of the influential factors was the foliar application of magnetic water, which had the greatest impact on morphological and physiological traits. Magnetic water improved plant growth by increasing nutrient solubility. The Ghaziantep landrace had the highest seed yield, biological yield, chlorophyll *a* and *b* contents, and N, P, and K contents. The Urmia landrace had the highest mean essential oil yield of both years treated with manure. Organic fertilizers improved plant growth and quality due to their effect on the availability of most micronutrients and macronutrients compared to chemical fertilizers. Furthermore, the Urmia landrace had the highest anethole percentage. In another study, it seems necessary to compare the Urmia landrace with the Ghaziantep landrace in the same or different treatment conditions to select the better one for sustainable production of medicinal plants and the replacement of chemical inputs.

**Author Contributions:** Methodology, formal analysis and writing; S.F., Formal analysis; M.T., Conceptualization, Supervision and project administration; R.A., Writing—Review and editing H.A.E.E. and R.Z.S. All authors have read and agreed to the published version of the manuscript.

**Funding:** This work was funded by Research Management Center (RMC), Universiti Teknologi Malaysia (UTM), through grant No. R.J130000.7344.4C240 and R.J130000.7609.4C359.

**Institutional Review Board Statement:** Not applicable.

**Informed Consent Statement:** Not applicable.

**Data Availability Statement:** All the data is available in the manuscript file.

**Acknowledgments:** We would like to thank Karel Innemie from Leiden University (The Netherlands), head of the Dutch archaeological mission at Wadi El-Natrun, who provided us data and some of the figures inserted in this study.

**Conflicts of Interest:** All the authors declare no conflict of interest.

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
