# Peer review of "The Effect of Foliar Application of Magnetic Water and Nano-Fertilizers on Phytochemical and Yield Characteristics of Fennel"

_horticulturae, doi:10.3390/horticulturae7110475_

Round 1
Reviewer 1 Report
Dear Authors
Please improve the manuscript especially in Results& Discussion, materials and methods. The manuscript must be revised by native English language.
Regards

Author Response
Comments |
Answer |
Reviewer 1 |
|
The (21) |
Corrected=line 24 |
By which treatmentØŸ (27)
|
Corrected=line 30 |
I could not see 77.2% (32) |
Corrected=line 36 |
9.66%)---- 9.61. (33) |
Corrected=line 36 |
2.15% ----- 2.18% (33)
|
Corrected=line 37 |
Yazad or Hamedan? (33)
|
Corrected=line 37 |
Please change the keywords (35) |
Corrected=line 42 |
Please also add this reference: (85)
|
Corrected=line 92 Bistgani, Z. E., Siadat, S. A., Bakhshandeh, A., Pirbalouti, A. G., Hashemi, M., Maggi, F., &Morshedloo, M. R. (2018). Application of combined fertilizers improves biomass, essential oil yield, aroma profile, and antioxidant properties of Thymus daenensisCelak. Industrial Crops and Products, 121, 434-440. |
Please apply SI unit. (ppm)(124) |
Corrected=line 132 |
Please add unit of EC (124) |
Corrected=line 132 |
N (%) (126) P (%) K (%)
Please report according to the mg/kg as the same cu, zn
|
Corrected=line 134 |
How much did you use? (133)
|
Corrected=line 141 |
0.2 g is really small to determine chlorophyll (155)
|
Corrected=line 168 Arnon, D.T. Copper enzymes in isolated chloroplasts polyphenol oxidase in Beta vulgaris. Plant. Physiol.1949, 24, 1-15. |
Please explain the extraction steps briefly and usefully (160)
|
Corrected=line 170
|
Please explain the extraction steps briefly and usefully (163)
|
Corrected=line 178
|
Please explain the extraction steps briefly and usefully (166)
|
Corrected=line 191
|
Please describe the steps of analysis of essential oil compounds completely. The specifications of the GC and GC-MS devices are fully described. (169)
|
Corrected=line 207
|
Which version? (175)
,
|
Corrected=line 223 (ver. 9.4) |
Which year? (189)
|
Corrected=line 239
|
Which year? (191)
|
Corrected=line 240
|
This sentence is not enough for discussion Please more discuss (193)
|
Corrected=line 243
|
You can also add below reference Influence of chitosan concentration on morpho-physiological traits, essential oil and phenolic content under different fertilizers application in Thymus daenensis. author(197) MasoudHashemiZohrehEmamiBistgani , Seyed Ata Siadat , AbdolmehdiBakhshandeh , AbdollahGhasemiPirbaluti
|
Corrected=line 250
EmamiBistgani, Z.; Siadat, S.A.; Bakhshandeh, A.; GhasemiPirbaluti, A.; Hashemi, M. Influence of chitosan concentration on morpho-physiological traits, essential oil and phenolic content under different fertilizers application in Thymus daenensis. J. Medi. Herb. 2016; 7, 117-125. |
Please discuss in detail how the metabolism of dry matter increases by organic fertilizers (199)
|
Corrected=line 251
|
You should note what percentage increase or decrease compared to the control treatment or other treatments. (212)
|
Corrected=line 270 |
Please describe the mechanism of how foliar application led to the increase (227)
|
Corrected=line 287 |
I see 12.3 (272)
|
Corrected=line 334
|
Are you sure about Table 6? (275)
|
Corrected=line 335 |
Please review again (277)
|
Corrected=line 339 |
Why did you say seed yield? (278)
|
Corrected=line 340 |
The sentence structure is incorrect (282)
|
Corrected=line 344 |
Are you talking about essential oil yield or essential oil percentage? (283)
|
I'm talking about essential oil yield and the following is the about essential oil. Cosge et al. [76]is deleted |
Please apply mg/gor mg/kg (295)
|
Corrected=line 356 |
The results of essential oil compounds are not enough at all, please explain in detail. (351)
|
Corrected=line 412 |
Please apply other words instead of that (353)
|
Corrected=line 414 |
Which Table? (354)
|
Corrected=line 417 |
Which Table? ((369)
|
Corrected=line 434
|
|
|

Reviewer 2 Report
The authors have demonstrated a comprehensive study on the effects of magnetic water and nano, organic, and chemical fertilizers on phytochemical and yield parameters of different landraces of the fennel plants. I have the following suggestions to enhance the scientific soundness and quality of the presentation of this article:
Please use uppercase for title, subtitles and headings, and subheadings. Also, I suggest the title be shortening as
The Effect of Foliar Application of Magnetic Water and Nano-fertilizers on Phytochemical and Yield Characteristics of Fennel
Line 17 to 21: Please rewrite this section, it is a very long sentence in the current format.
Line 27 and 28: This is so general and unclear, please be specific that the mentioned parameters increased after foliar application of all treatments compared to the non-treated check?
Line 34: Author(s) should add a straightforward takeaway message on the future perspective of these nano-fertilizers as sustainable plant growth promoters.
Line 35-36 please use ";" to separate the keywords instead of ","
Also, again the keywords are too long example “systematic literature mapping of self-adaptive system test”
Line 38: Suggestion: Just use “medicinal plants”
line 106: Please edit as follow: comparison with the local landrace (Urmia) of Fennel.
Please edit Line 210: 3.2. Fruit Yield and Biological Yield
Line 232: years.
Line 386-387: Better to avoid repeated use of the same word in the same paragraph e.g. influential
Line 384: Please answer the following questions in the conclusion section:
what are the directions for the future? 
what are the research gaps?
what is the novelty or what’s new to this article?
Line 399: what is RMC? Please add the full name
Line 403: Please reformat the references according to the journal guidelines
Examples:
The journal names should be abbreviated and in italic
the years of articles should be bold also please add the DOI link if available for each reference

Author Response
Reviewer 2 |
|
Please use uppercase for title, subtitles and headings, and subheadings. Also, I suggest the title be shortening as
|
Corrected=line 1 |
Line 17 to 21: Please rewrite this section, it is a very long sentence in the current format.
|
Corrected=line 19 |
Line 27 and 28: This is so general and unclear, please be specific that the mentioned parameters increased after foliar application of all treatments compared to the non-treated check?
|
Corrected=line 30
|
Line 34: Author(s) should add a straightforward takeaway message on the future perspective of these nano-fertilizers as sustainable plant growth promoters.
|
Corrected=line 37
|
Line 35-36 please use ";" to separate the keywords instead of "," Also, again the keywords are too long example “systematic literature mapping of self-adaptive system test”
|
Corrected=line 42
|
Line 38: Suggestion: Just use “medicinal plants”
|
Corrected=line 44
|
line 106: Please edit as follow: comparison with the local landrace (Urmia) of Fennel.
|
Corrected=line 114 |
Please edit Line 210: 3.2. Fruit Yield and Biological Yield
|
Corrected=line 268 |
Line 232: years.
|
Corrected=line 293
|
Line 386-387: Better to avoid repeated use of the same word in the same paragraph e.g. influential
|
Corrected=line 453 |
Line 384: Please answer the following questions in the conclusion section:
|
1-The aim to identify high- yielding compatible ecotypes and introduce them to farmers for alternative planting 2- There was no particular problem and all ecotypes showed relative compatibility 3- Use of foliar spraying of magnetic water and comparison of several ecotypes, especially Turkish ecotype with Iranian ecotypes. |
Line 399: what is RMC? Please add the full name
The authors would like to thank RMC |
Corrected=line 465 |
Line 403: Please reformat the references according to the journal guidelines
|
Corrected |

Round 2
Reviewer 2 Report
The quality of this manuscript is greatly enhanced after revision and I only have some minor comments on keywords & tables that I add as a note.
You may use a regular font instead of bold
The keywords, you may use lower case
Tables heading, please use . at the end of each heading

Author Response
Reviewer 2 Round 2 Report
Comments and Suggestions for Authors
The quality of this manuscript is greatly enhanced after revision and I only have some minor comments on keywords & tables that I add as a note.
- You may use a regular font instead of bold
Authors response: Agreed and corrected in Tables 2, 3, 11 and 12
- The keywords, you may use lower case
Authors response: Agreed and corrected. Line 41
- Tables heading, please use . at the end of each heading
Authors response: Full stop added at the end of headings of all the Tables.